# An Authoritative Study on the Near Future Effect of Artificial Intelligence on Project Management Knowledge Areas

**Thordur Vikingur Fridgeirsson \*, Helgi Thor Ingason** 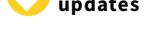**, Haukur Ingi Jonasson and Hildur Jonsdottir**

Department of Engineering, Reykjavik University, 101 Reykjavík, Iceland; helgithor@ru.is (H.T.I.);
haukuringi@ru.is (H.I.J.); hildurj6@ru.is (H.J.)
**\*** Correspondence: thordurv@ru.is

**Abstract:** The purpose of this study is to explore how Artificial Intelligence (AI) might augment the project management profession in each of the 10 categories of project management knowledge areas, as defined in the Project Management Body of Knowledge (PMBOK) of the Project Management Institute (PMI). In a survey, a group of project management experts were asked to state their insights into AI's likely effect on project management in the next 10 years. Results clearly illustrated that AI will be an integrated part of future project management practice and will affect project management knowledge areas in the near future. According to these findings, the management of cost, schedule, and risk, in particular, will be highly affected by AI. The research indicates that AI is very useful for processes where historical data is available and can be used for estimation and planning. In addition, it is clear that AI can monitor schedules, adjust forecasts, and maintain baselines. According to the findings, AI will have less impact in knowledge areas and processes that require human leadership skills, such as developing and managing teams and the management of stakeholders. The results indicate proprietarily the project management knowledge areas as defined by PMI that AI is likely to augment and sustain.

**Keywords:** project management; artificial intelligence; PMBoK knowledge areas; project manager

## 1. Introduction

Alan Turing (1912–1954) laid the foundation for the popular understanding of how Artificial Intelligence (AI) should function as a model of human calculations rather than of pure mathematical calculi [1]. Turing put forward the notion that certain functions cannot easily be emulated by computers. The famous Turing Test method of inquiry for determining whether or not a computer is capable of thinking like a human being is rarely passed by a computer [2]. The earliest definition we can find of AI is, however, attributed to McCarthy, Minsky, Rochester, and Shannon [3], who described it as an action performed by a machine that would be considered intelligent if done by a human. In 1956, Newell and Simon predicted that AI would be essential for management. Since then, the expected impact of AI on management has been discussed by many researchers, such as Kolbjørnsrud, Amico, and Thomas [4] who discussed how AI will redefine management and save a lot of time spent on administrative coordination and control. Heukamp and Canals [5] presented 10 ways in which AI is transforming management—and to which managers need to pay attention, in order not to become complacent, and Noponen [6] discussed the impact of AI systems on management during the next decade and concluded that AI can complement human decision making, particularly for managers at the highest levels of organizations. [7] claim that AI is primarily investigated in computer science and operation research for a deeper understanding of how automation will shape our future, but to a lesser extent on how humans can collaborate with machines to perform a task—in short, how AI can not only automate processes but also augment the management domain. A survey conducted by the Accenture Institute for High Performance among 1770 managers on different levels in 14 countries indicates that middle managers, such

as project managers, are significantly more skeptical about taking advice from intelligent systems than higher ranked managers [4].

Researchers who have studied the future impact of AI [8–11] estimated that AI is likely to increase the productivity of some workers and replace others, but what fields will be most affected remains unknown. The importance of projects in modern societies has been defined in fiscal terms by Schoper, Wald, Ingason and Fridgeirsson [12] who demonstrated that the share of project-related work in advanced economies is one third, and rising. The Schoper et al. [12] study indicates that the relative share of projects of the Gross Value Added to the economy of Germany, Norway, and Iceland in the period of 2009 to 2019 escalated more than 20%. This fact puts the importance of project management into context, and we argue that the sustainable development of any organization is very much linked to its capability to deal with change. AI is likely to shape the discipline of project management and thus play a role in this development in the near future. Best practices, education and training, economical metrics and the open system theories, that govern the scientific platform of project management, are examples of professional challenges that awaits the discipline.

It is essential to gain an understanding of the areas of the project management knowledge base that will require most attention from the perspective of the profession itself in the context of AI. Each specialized profession requires different skills, and an understanding of the specificity of the project management model for work-related tasks and skills could make it easier to predict what effect AI will have on the project management profession [13]. Yet, only rather limited published research can be found on this aspect, as we discuss below. The authors believe that this study will indicate opportunities and benefits on how machine learning and AI can contribute to sustainability. The study isolates the managerial knowledge areas to enhance the projectized business model and predicts the organizational transformation that is awaiting for management practitioners in the near future.

The aim of this research is to obtain an overview of the project management attributes that are best amplified by AI, according to project management experts, and thereby contribute to filling the research gap that we point out. The research model is based on the 10 project management knowledge areas from the Project Management Body of Knowledge (PMBOK) published by The Project Management Institute (PMI). Other noteworthy organizations that contribute to project management competence baselines are the International Project Management Association (IPMA) and the Association for Project Management (APM). PMBOK was selected as the research platform because of how well structured, comprehensive and accessible the guidelines are for the purpose. Furthermore, PMBOK is used worldwide as a reference for best practices in the project management profession [14]. The authors are aware that the PMI standards are based on empirical evidence from the industry rather than on scientific work. However, the PMBOK standard serves excellently as a platform for systematic studies on how the managerial knowledge areas will be impacted and paves the way for further research. The questionnaire is designed from 49 processes of the 10 project management knowledge areas and has 53 questions. There are four general questions about the participants' backgrounds, followed by 49 questions on what they deem to be the likely effect they think AI will have on each process in the next 10 years.

For the purpose of this work, the 10 knowledge areas of the PMBOK are used as a descriptive reference for the knowledge base for project management. The basic research question for this work is: what aspects of project management, as defined by the Project Management Body of Knowledge, will be affected by artificial intelligence in the next 10 years? Detailed speculations about the essence and extent of this impact are, however, not within the scope of this paper.

## 2. Background

### 2.1. Project Management Knowledge Base

The definition and understanding of the role of a project manager has been changing over the years since this concept was introduced for the first time by Gaddis [15]. Research

shows that project managers look at their tasks and assignments differently [16,17]. However, the knowledge areas of project management have been described by international project management associations, such as the Project Management Institute (PMI) and the International Project Management Association (IPMA). These are non-profit associations and certification bodies for the field of project management. The IPMA is an association of national project management associations and maintains its Individual Competence Baseline (ICB), which is now available in its fourth edition. The IPMA defines the competence areas for project managers in terms of people (interactions with oneself and other people), practice (related to management of projects, programs and portfolios) and perspective (the context within which a project is run) [18]. The PMI serves around 2.9 million project management specialists who are located all over the world (PMI, n.d.) [19]. PMI volunteers with diverse experience have developed and maintain up-to-date, globally recognized standards for project, program, and portfolio management. The PMI has maintained its Guide to the Project Management Body of Knowledge (PMBOK) since 1996, and its most recent version (sixth edition) was published in 2017. The PMBOK is a professional guide that offers guidelines for the practice of project management. The PMBOK Guide describes 49 processes that are categorized by 10 knowledge areas. The 49 processes are divided logically into five project management process groups that aim to achieve defined objectives: (i) initiating, (ii) planning, (iii) executing, (iv) monitoring and controlling and (v) closing. The 10 knowledge areas are: (i) integration, (ii) scope, (iii) schedule, (iv) cost, (v) quality, (vi) resources, (vii) communications, (viii) risk, (ix) procurement and (x) stakeholders. Each knowledge area is defined by its knowledge requirements. As an example of this, the knowledge area "integration" is supported by; developing project charter, project management plan, project work, knowledge, monitoring, change control, and project closing, etc.

### 2.2. Artificial Intelligence and Project Management

For the purpose of this paper, artificial intelligence (AI) is defined as a system's ability to accurately interpret external data, learn from the data, and use what it learns to complete specific goals and tasks [20]. In their 2010 book Artificial Intelligence: A Modern Approach, Russel and Norvig talk about four approaches to AI: (i) acting humanly, (ii) thinking humanly, (iii) thinking rationally, and iv) acting rationally. AI technology is founded on building a knowledge base that is not programmed directly, but forms and accumulates as the machine "learns" in a designed environment. Processes that utilize data and use artificial intelligence technologies can then be turned into automated processes [21].

Researchers have come to different conclusions on how much impact AI will have on future jobs. To some extent, this depends on the research method used. Frey and Osborne [8] were two of the first to publish research about AI's effect on the future of labor. They concluded that 47% of jobs in the USA were at a high risk of becoming automated. Research done in the 32 OECD countries looked at automation impact more precisely by looking at individual jobs, not occupations [10]. The research concluded that half of job occupations would most likely change considerably because of automation. The effect of these changes, however, is quite widespread. About 14% of jobs in the OECD countries are at risk of becoming 70% automated. About 32% of jobs are at risk of moderate change by automation, with automation of 50 to 70%. Another research estimated a 35% risk to employment due to job automation in Finland and 33% in Norway [11] The OECD study also showed that the higher the level of education needed for the job is, the less likely the job is to be automated, at least in the short term. The same applies to income—the higher the income, the lower the probability of automation [10].

In the short run, AI may reshape the demand for specific working skills in the labor market, mainly because it generally performs specific tasks. These small changes to the demand might, however, accumulate into more drastic changes in the labor market; changes including redefinition of occupational skills, job creation and increased unemployment in technical fields [13]. The skill requirement for jobs is not a constant and can develop and change over time. Certain skills, such as social skills, are difficult to automate [13].

Raich and Krakowski [22]. reviewed three popular, recent business books on the use of AI in organizations and concluded that the essence of what they say is that AI should correspondingly be used to automate processes and augment managerial functions.

AI in project management is not a totally new subject. Alden T. Foster [23]. stated that AI could be successfully applied to project management so as to analyze large datasets to find patterns, trends, and problems that need attention, based on previous knowledge. AI could also monitor how the project is going and make changes to future activities if needed. Project management schedules that are rule-based are ideal for AI methods, resource-constrained scheduling and time, cost, and risk schedule [23]. The process of building a project network could be made faster and more productive with the help of AI. With the help of AI, project management tasks can be done automatically, and AI can direct a project and help make related decisions [24].

Three organizations—Arup, The Bartlett School of Construction and Project Management at University College London (UCL), and the Association for Project Management (APM)—collaborated and published the report, Future of Project Management, in 2017 [25]. The report is a discussion about future changes in the project management profession and it is divided into seven different trends. One of the trends concerns automation and human-machine collaboration. The report indicates that intuitive user interfaces will cause fundamental changes to workplaces, collaboration and communication and that it could also help in creating and maintaining a good atmosphere for improved productivity and creativity in the workforce.

Authors of an article from PricewaterhouseCoopers are convinced that AI will transform how project management processes will be delivered and managed in the near future [26]. AI will evolve from simple task automation to predictive project analytics, advice and actions.

A variety of research is available on the use of AI methods for specific project management tasks, such as project forecasting, project cost, and schedule success [27–29]. and also research on hybrid systems based on AI methods that have performed well in cost estimation and project risk calculations [30,31].

The main obstacle preventing the use of AI in project management is the lack of knowledge and understanding of AI and also the time and cost of implementing an AI system [23]. AI needs large historical datasets and project information in a standardized form to work with, which can be challenging to provide [26]. Research also shows that managers are more willing to use and put their trust in an AI system if they understand how it works and how it can generate advice, and also if the system provides convincing explanations and has a proven track record [4].

Even though AI will apparently have a high impact on project management, AI is not human and there will still be a need for flesh and blood project managers. Human skills such as empathy, emotional intelligence, negotiation, decision-making, and human resource management will be valuable in the near future—perhaps more than ever [25,26].

PMI acknowledges that project managers will need digital skills to keep up with the emerging technology [32]. When looking at the PMI talent triangle, the technical project management skill group is considered to have the most potential to be supported by AI functions. Here, AI project management bots, assistants, and algorithms can support project managers in their day-to-day work by analyzing the project status and providing observations and predictions with data. In strategic and business management, AI can support project managers by fitting parameters and making forecasts. In general, leadership might not be ideal for consideration in AI systems, but it could be in the future. It could, for example, provide a ranked list of candidates for a team based on requirements [26]. By using AI for projects, project managers can have more assistance, efficiency, insight, and strategy, resulting in more productivity [24].

If organizations and project managers are vigilant regarding available and appropriate AI systems to use, it might certainly make a big difference in value delivery [19]. Project managers can benefit from using AI systems to estimate cost, handle schedules, monitor

progress, reminders, follow-ups, and manage activities and plan resources [19,30,33]. With the application/applications of AI, project managers have extra time, which allows them to focus on team members and more specific and complex tasks that add more value [19,33].

To sum up, the literature review has indicated connections between the proposed impact of AI on the different project management competence areas, as follows:

## 3. Method

The purpose of this research is to get a project management specialist perspective on AI's future effect on project management, in order to get an understanding of what knowledge areas of project management will most likely be affected by AI in the foreseeable future. The unit of analysis for this research is the discipline of project management, as defined by the 10 knowledge areas of the PMBOK. The research was exploratory, since it investigates a topic on which little data is available and where there is still a lack of scientific research and theoretical literature. The study was also prospective, as it focused on the possible effect that AI could have on project management in the foreseeable future [34].

The method used for the study was a quantitative cross-sectional study. The quantitative method follows standardized forms that seek to quantify the results of the research and are represented analytically. A cross-sectional study is a research design where data is collected at one point in time, so as to get an overview of the study [34].

### 3.1. Research Design

The research was designed in accordance with the framework of the PMBOK. A convenience sample was used to gather data for the research. The sample of experts was defined, based on their educational background and the likelihood of being able to reach out to them and commit them to take part in the survey. For the above reasons, the sample chosen consisted of individuals who had graduated with a Masters in Project Management (MPM) from one of two Icelandic universities between 2007 and 2019. These people were suitable for the survey due to their knowledge and work experience and because they were also easy to access. The structure and objectives of this program has been described by Ingason and Jonasson [35]. Some fundamental assumptions can be made regarding the knowledge, skills and competences of this group in terms of basic understanding of project management. Such assumptions are a consequence of the admission requirements for the program, the compendium covered during the program and the practical experience gained by the students through the program.

The questionnaire consisted of 53 questions, divided into two parts. In the first part were background questions and in the second part were questions about the predicted effect of AI in the context of the 10 project management knowledge areas of PMBOK. The background questions covered gender, age, project management certifications, and knowledge of the 10 project management knowledge areas. The questions on the possible impact of AI were 49 in number and their order was random. One question was asked for each of the processes of the 10 project management knowledge areas. The survey asked how much effect the participants believed AI would have on each process of the 10 project management knowledge areas in the near future.

Closed questions were used with predetermined answer options for all of the questions, except for one background question. In the question on project management certification, "another" choice was available, which offered an opportunity to write an answer. For the questions in the second part of the survey, an ordinal scaling was used with the following options: "very low effect," "low effect," "medium effect," "high effect," and "very high effect." An option to answer with "don't know" was also given.

The questionnaire was conducted through SurveyMonkey. When an online survey is used for collecting data, the participants must understand the aim and relevance of the research [34]. With that in mind, an email was sent out to 395 participants on 9 April 2020, with information on the aim of the research, confidentiality and a link to

access the questions. When the survey was closed on 29 April 2020, 81 complete answers had been received.

The answers were processed in Excel and SPSS. In SPSS, the main questions were changed into binary variables. The "don't know" option was given the value zero, and the other ranking options were combined and given the value one. This was done to analyze whether there was an association between the background variables and when the respondents did not know how much effect AI would have on the 10 knowledge areas. Contingency tables and chi-square tests in SPSS were used to check the relationship between the respondent's background variables and binary variables. The chi-square test checks the relationship between nominal variables. Pearson's chi-square test was used to test for an association between gender and the binary variables, and Fisher's exact test for the other background variables and the binary variable. Phi ($\varphi$) gave the strength of the association for gender and the binary variable [35]. Fisher's exact test is used when the sample size is small, and assumptions for the Pearson's chi-square test are not met. Fisher's exact test calculates the exact probability of the chi-square statistic (Fisher, 1922 in Field, 2009).

The "don't know" option was excluded from the main questions before an association, and correlation was examined for the other options. The assumptions for the Pearson's chi-square test were not met, so Fisher's exact test was used to test for associations between gender, respondent's certification, and the predicted effect the respondents thought AI would have on the processes of the 10 knowledge areas.

Spearman's rho ($r_s$) correlation was used to analyze the correlation between the respondent's age and knowledge of the 10 project management knowledge areas to the responses to the main questions of the questionnaire. Spearman's rho correlation is a nonparametric rank statistic that calculates the intensity of the association between ordinal or ranking scale variables. Correlation and assumptions are significant if the significance (p) is less than 0.05 [35].

### 3.2. Quality of Research

Validity in research has to do with the appropriateness and accuracy of the research instrument used to measures a given reality. The design of the questionnaire was conducted in such a fashion that it would give sound results and accurately answer the research question. The 10 project management knowledge areas from the PMBOK were used as the research framework. The research is considered valid because each question had a logical link with an objective and the questionnaire covered all aspects needed.

Ethical considerations in this research had to do with informed and unrestricted consent from participants [36]. Participants in the survey received a link via email and thus had full freedom to refuse or discontinue participation at any time. They were also informed about the purpose of the research and how the results would be used, as well as being provided with a statement of confidentiality.

In retrospect, in the middle of the Covid19 pandemic, one could also praise the fact that an online survey was used to collect the data. Questionnaires also have their disadvantages, which include the risk of a low response rate. It is also not possible to know under what circumstances the questionnaire was answered, and it is not easy to determine what factors might affect the responses. The understanding of the questions may also differ between participants, and there is a lack of opportunity to clarify misunderstanding and other issues [36].

### 4. Results

The background information on those who answered the survey was reviewed. A total of 81 respondents, all with a background and extensive working experience in PM, and with a Master of Project Management (MPM) degree answered the survey, 39 females and 42 males. The gender distribution was relatively equal between age groups, except in the 60-years-old or older group. In that group there were seven males and one female.

Only two of the 81 respondents did not hold an international project management certification. Most of the participants, or 76.55% (62), claimed that they knew the PMBOK knowledge areas quite well or very well, and one individual claimed to know it extremely well. Only 2.47% (2) said that they did not know the 10 knowledge areas "at all", and 19.75% (16) "not so well".

### 4.1. AI Effect on the 10 Knowledge Areas

The 10 project management knowledge areas of PMBOK contain several processes, each area from three to seven processes. Respondents answered how much effect they believed AI would have on each of the processes over the next 10 years. In order to obtain overall results for the 10 PMBOK knowledge areas, the average scoring percentage of processes in each knowledge area was found.

The average scoring result shows that the participants believe that AI will likely have the highest effect on project cost management, 58% thought AI would have a very high or high effect on project cost management. 51% of respondents thought AI would have a very high or high effect on project schedule management, and 47% on project risk management in the next 10 years. The respondents believe that AI would have a medium to high effect on project quality management or 61%, and 60% on project procurement management. According to the results, AI would likely have the least effect on project stakeholder management in the next 10 years. For the other areas of knowledge, project integration management, project scope management, project resource management, project communication management, the result was very similar. The distribution of the responses was divided between low effect, medium effect, and high effect. All percentages can be seen in Table 1 and further analyses in the next sections of this chapter.

**Table 1.** Examples of publications on AI effect in context of the 10 knowledge areas.

| Knowledge Areas: | Example of Publications: |
| --- | --- |
| Project Integration Management | [24] |
| Project Scope Management | [34] |
| Project Schedule Management | [19,23,27,28,28,29,29,30,34] |
| Project Cost Management | [19,27–30,34] |
| Project Quality Management | [36] |
| Project Resource Management | [19,30,34] |
| Project Communication Management | [33,37] |
| Project Risk Management | [23] |
| Project Procurement Management | |
| Project Stakeholder Management | [26] |

Table 2 shows an overview on the effect on managing the respective knowledge areas in the near future.

**Table 2.** AI effect on the 10 knowledge areas in the next 10 years.

| The PMBOK Knowledge Areas | | The Effect on Managing the Knowledge Areas | | | | | |
|---|---|---|---|---|---|---|---|
| | | Don't know | Very low | Low | Medium | High | Very high |
| 1 | Integration | 14% | 4% | 19% | 26% | 30% | 7% |
| 2 | Scope | 15% | 3% | 23% | 26% | 27% | 6% |
| 3 | Schedule | 16% | 0% | 10% | 23% | 37% | 14% |
| 4 | Cost | 13% | 1% | 5% | 23% | 40% | 18% |
| 5 | Quality | 14% | 3% | 14% | 31% | 30% | 8% |
| 6 | Resource | 14% | 5% | 21% | 29% | 24% | 7% |
| 7 | Communication | 14% | 4% | 22% | 29% | 25% | 6% |
| 8 | Risk | 15% | 1% | 13% | 24% | 35% | 12% |
| 9 | Procurement | 18% | 1% | 13% | 31% | 28% | 9% |
| 10 | Stakeholder | 16% | 9% | 27% | 27% | 18% | 3% |

*4.2. AI Effect on Project Integration Management*

The participants believe that AI will have the most effect on monitoring and controlling project work, 49% thought it would have a very high or high effect. It was also assumed that AI would have a high effect on developing a project management plan, developing a project charter, and performing an integrated change control. AI was believed to have the least effect on managing project knowledge, closing a project or phase, and directing and managing project work. The main results for the processes of the project integration management can be seen in Figure 1.

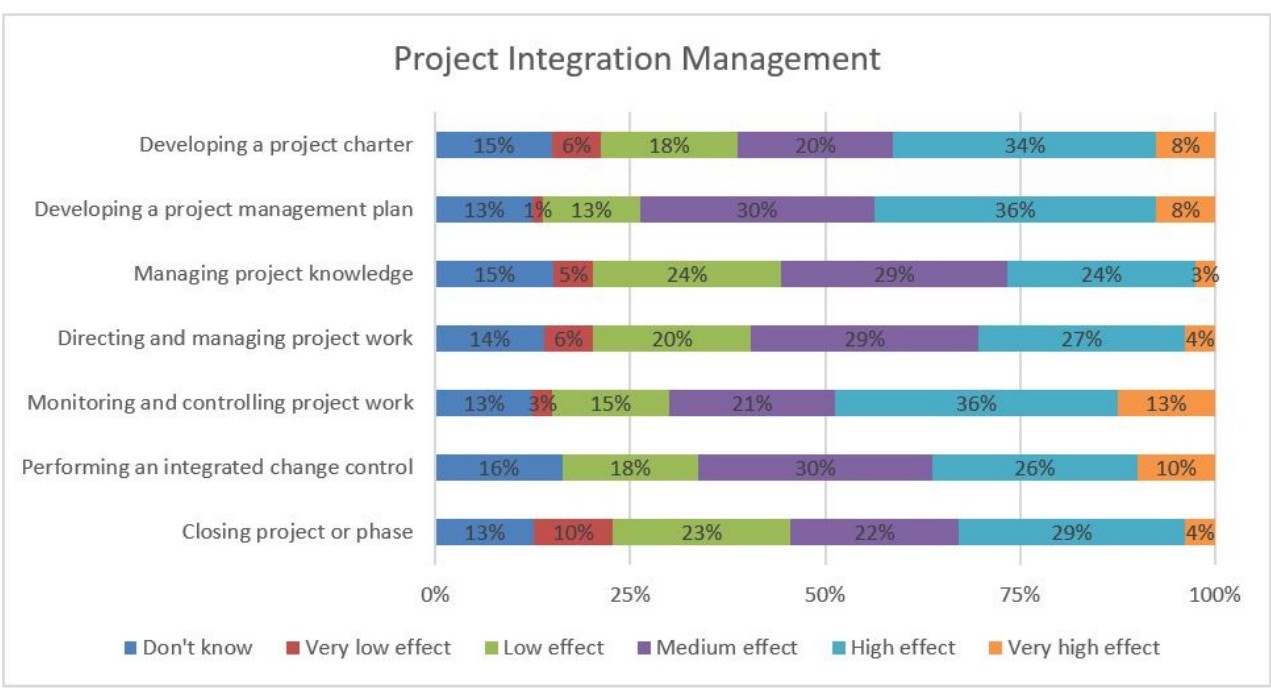

**Figure 1.** AI effect on processes of the Project Integration Management. The sum of the percentages should be 100%.

There was an association between gender and how much effect respondents thought AI would have on integrated change control, $p = 0.044$. Males were more likely to think that AI would have a low or medium effect on integrated change control, whereas females believed it would have a high or very high effect. Of those who said AI would have a low effect on integrated change control, 90.9% were males.

The age of the respondents did not have any correlation to how they answered the questions about AI's effect on processes of the project integration management. There was a significant correlation between how well respondents knew the 10 knowledge areas and the effect they believed AI would have on integrating change control, rs = 0.416, and $p$ = 0.003. The more knowledge of the 10 knowledge areas they had, the higher effect they thought AI would have on integrating change control. There was no correlation between the respondents' knowledge and other processes of the project integration management.

### 4.3. AI Effect on Project Scope Management

AI is likely to have the greatest effect on creating a work breakdown structure (WBS), 50% answered high or very high effect. 40% of participants said that AI would have a very low or low effect on defining scope, 31% on controlling scope, and 27% on validating scope, see Figure 2.

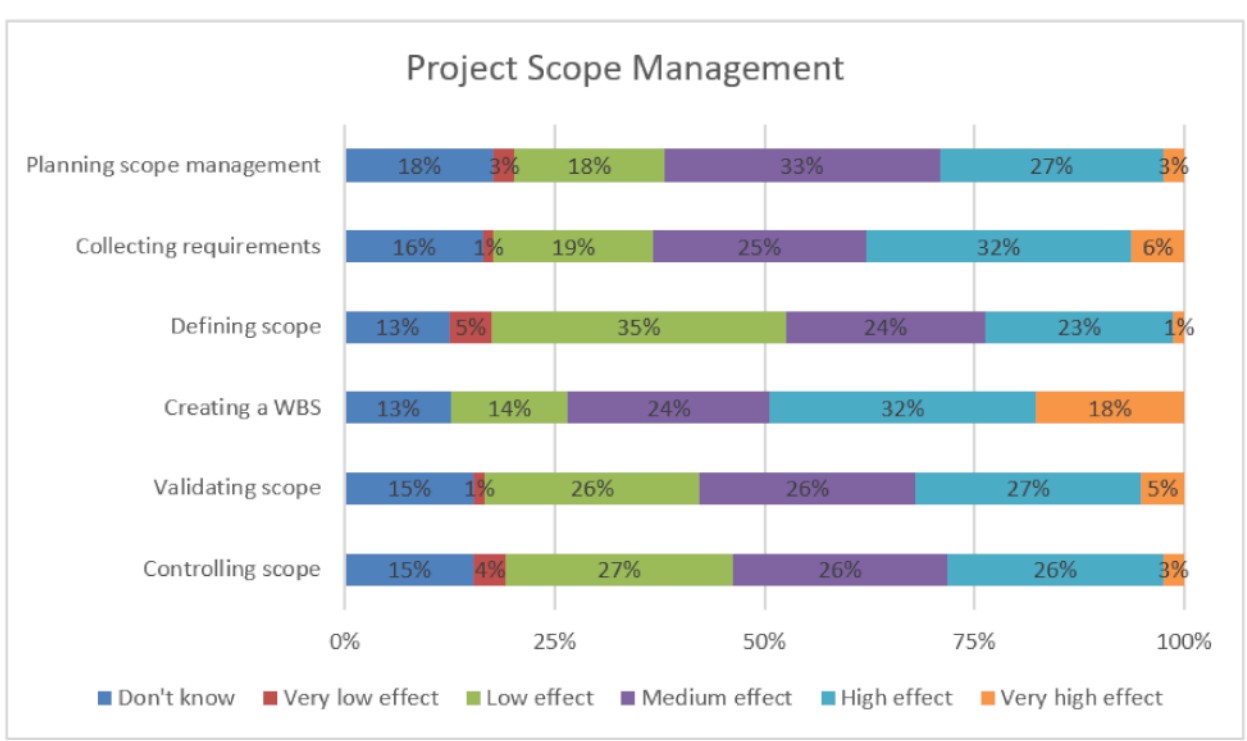

**Figure 2.** AI effect on processes of the Project Scope Management.

There was no association between either gender or certification level and how much effect respondents believed AI would have on processes of the project scope management. There was not a significant correlation between the participant's age or how well they knew the 10 project management knowledge areas and the answers to the effect on project scope management.

### 4.4. AI Effect on Project Schedule Management

Figure 3 shows that the results for all processes in project schedule management were somewhat equivalent. AI would have a very high or high effect in 40 to 56% of all the processes. The highest effect would be on the controlling schedule and the lowest effect on defining activities.

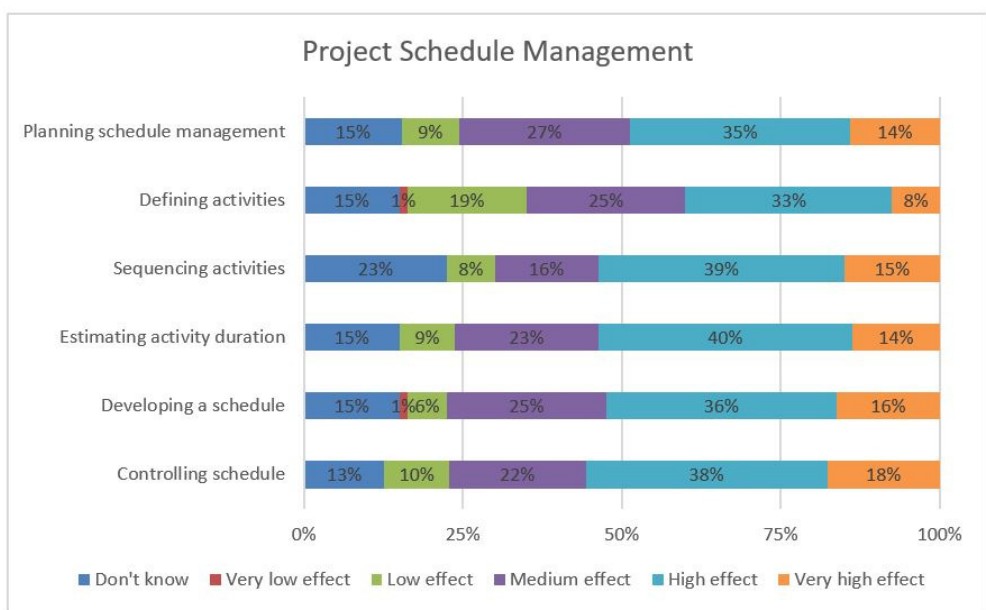

**Figure 3.** AI effect on processes of the Project Schedule Management.

There was no association between either gender or certification and how much effect respondents believed AI would have on processes of the project schedule management.

There was a significant correlation between how much effect respondents think AI would have on developing schedule and their age, rs = 0.312, *p* = 0.027. The higher the respondent's age, the more effect they believed AI would have on developing a schedule. 50% of those aged 60 and older thought that AI would have a high effect on developing a schedule and 40% in the 50-to-59-year age group. There was a significant correlation between respondents' knowledge of the ten knowledge areas and both estimating activity duration, rs = 283, *p* = 0.047, and controlling schedule, rs = 0.329, *p* = 0.021. The more knowledge respondents had on the ten knowledge areas, the more effect they thought AI would have on both estimation activity duration and controlling schedule. Of all those who said that AI would have a high effect on estimating activity duration, 70% knew the ten knowledge areas very well. The same applied for the controlling schedule, 63.6% knew it very well.

### 4.5. AI Effect on Project Cost Management

High effects were found when processes of project cost management were analyzed. This showed that 54 to 64% thought that AI would have a very high or high effect on the processes. Estimating costs showed the highest result of 64%. The results can be seen in Figure 4.

There was not a significant correlation between how much effect respondents thought AI would have on processes of the project cost management and their age.

There was a significant correlation between how much effect respondents thought AI would have on controlling cost and how well the respondents knew the 10 knowledge areas, rs = 0.356 and *p* = 0.011. The more knowledge respondents had on the 10 knowledge areas, the more effect they thought AI would have on controlling cost. The one person that knew the 10 knowledge areas extremely well thought that AI would have a very high effect on controlling cost, as did 43.8% of the respondents that knew the knowledge areas very well. There was not a significant correlation between how much effect respondents thought AI would have on processes of the project cost management and their gender, age, certification, or how well they knew the knowledge areas.

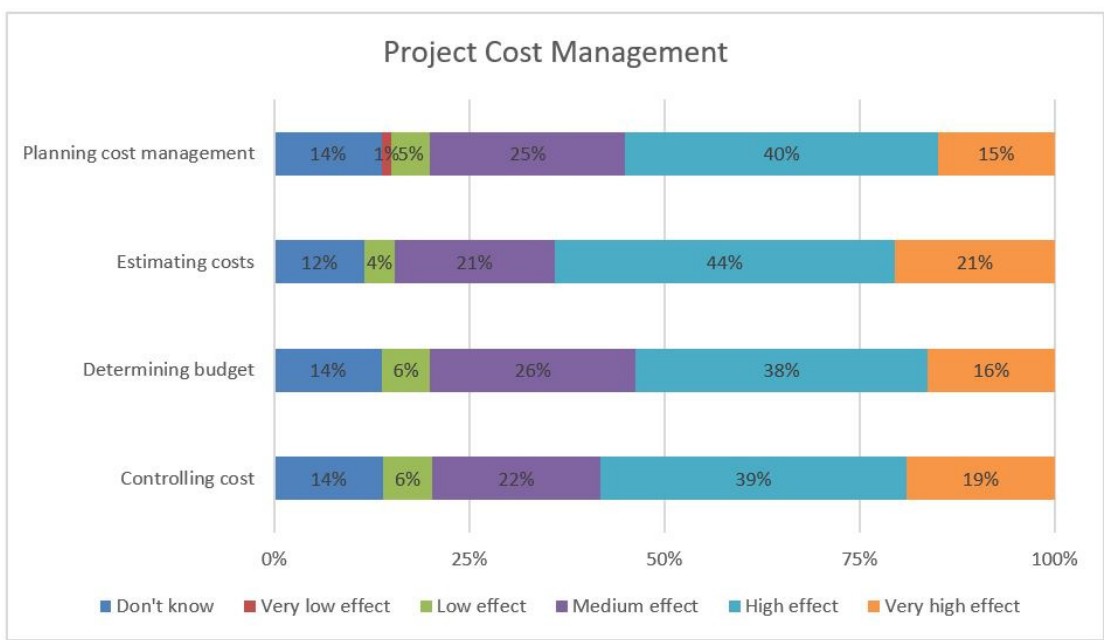

**Figure 4.** AI effect on processes of the Project Cost Management.

### 4.6. AI Effect on Project Quality Management

When reviewing the results for the project quality management, it can be seen that the results were almost identical for the three processes. Out of the participants, 38 to 39% said that AI would have a very high or high effect on the three processes and 30 to 32% a medium effect, see Figure 5.

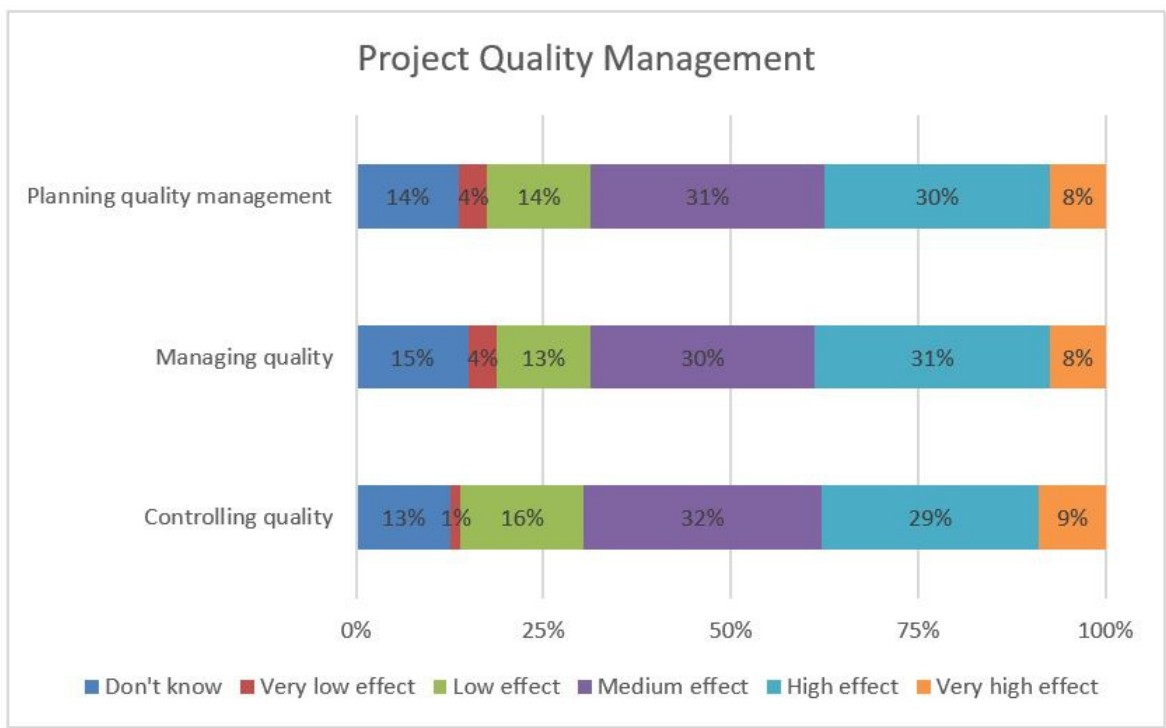

**Figure 5.** AI effect on processes of the Project Quality Management.

There was no association between either gender or certification level and how much effect respondents believed AI would have on processes of the project quality management. There was no significant correlation between the participants' ages or how well they knew

the 10 project management knowledge areas and the answers to the effect on the project quality management.

### 4.7. AI Effect on Project Resource Management

Figure 6 shows that AI was believed to have a low effect on managing and developing a team. Out of the participants, 14% said that AI would have a very low effect on managing a team, and 38% a low effect. For the process developing team, 13% answered that AI would have a very low effect and 31% a low effect. AI will have a low to medium effect on the process of acquiring resources. The result showed that regarding project resource management, the processes that AI would most likely affect to a high to moderate degree are planning resource management, controlling resources, and estimating activity resources.

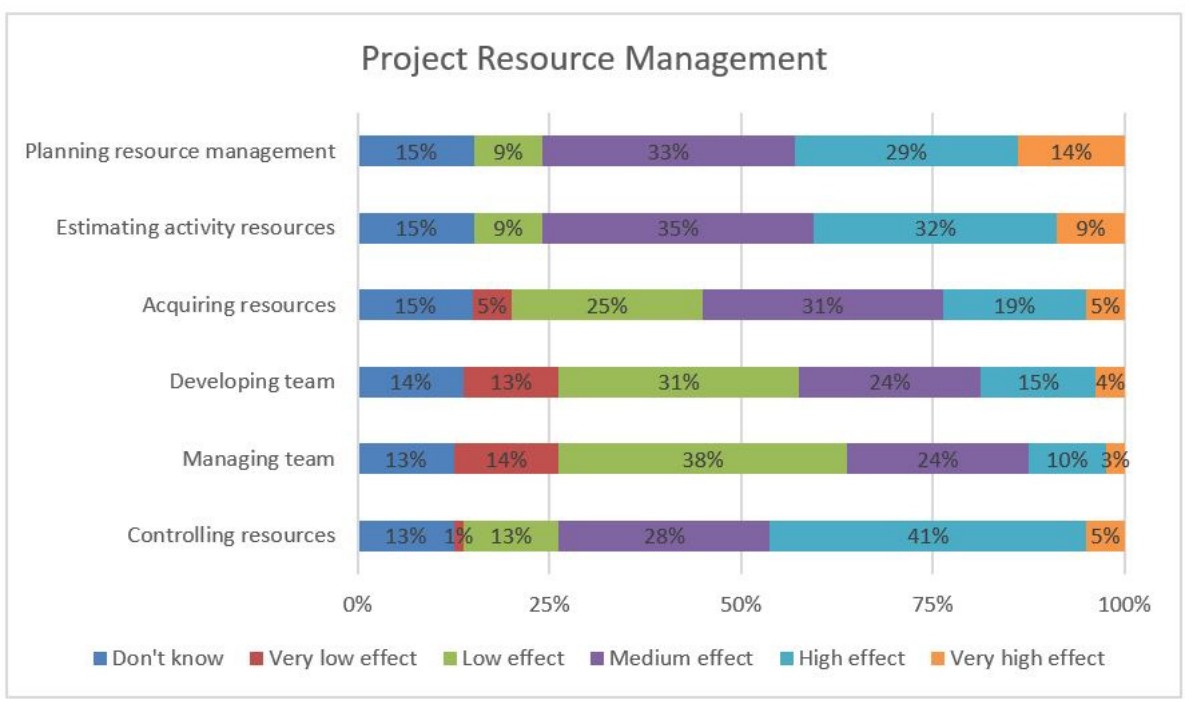

**Figure 6.** AI effect on processes of the Project Resource Management.

### 4.8. AI Effect on Project Communication Management

The results showed that AI would most likely have a low effect on planning communication management and managing communication. Out of the participants, 33% thought AI would have a very low to low effect on managing communication and 30% on planning communication management. Participants estimated that AI would have a medium to high effect on monitoring communication, as shown in Figure 7.

There was no association between either gender or certification level and how much effect respondents believed AI would have on processes of the project communication management. There was not a significant correlation between the participants' gender or how well they knew the 10 project management knowledge areas and the answers to the effect on the project communication management.

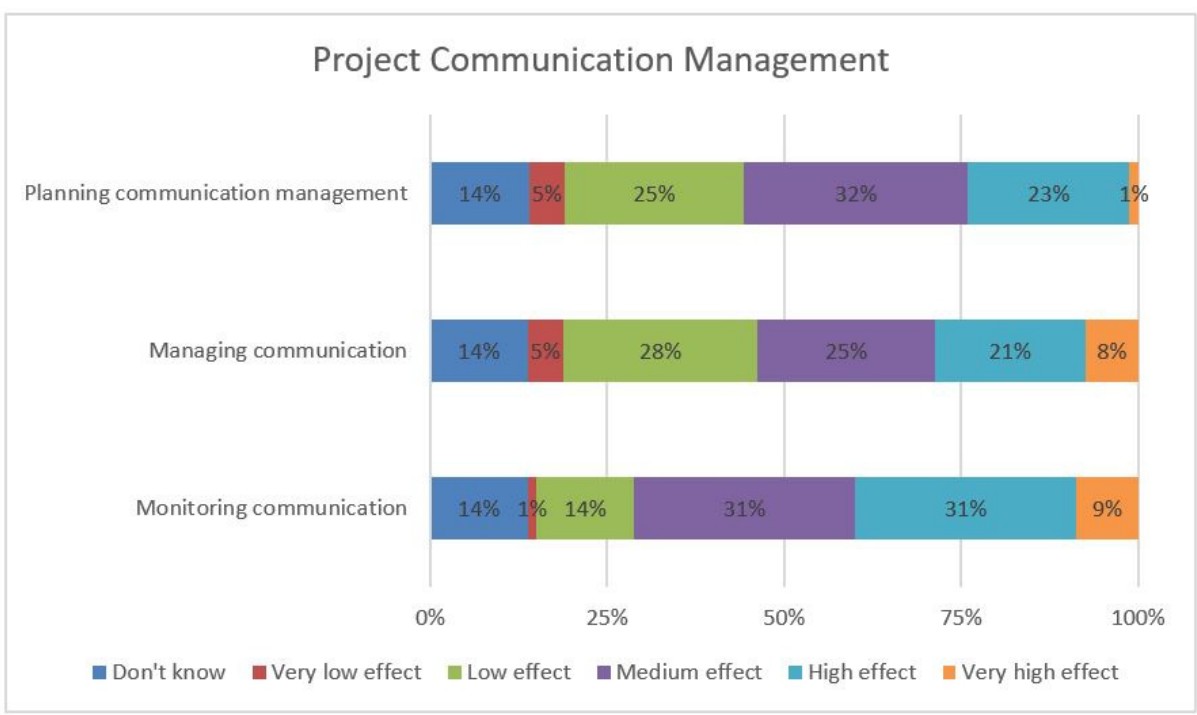

**Figure 7.** AI effect on processes of the Project Communication Management.

### 4.9. AI Effect on Project Risk Management

When the project risk management knowledge area was viewed, it showed that AI would most likely have a high effect on the processes of project risk management. Out of the participants, 63% believed that AI would have a very high or high effect on monitoring risks and 54% on performing quantitative risk analysis. The result shows that AI was believed to have the lowest effect on planning and implementing a risk response. The main results for the processes in the project integration management can be seen in Figure 8.

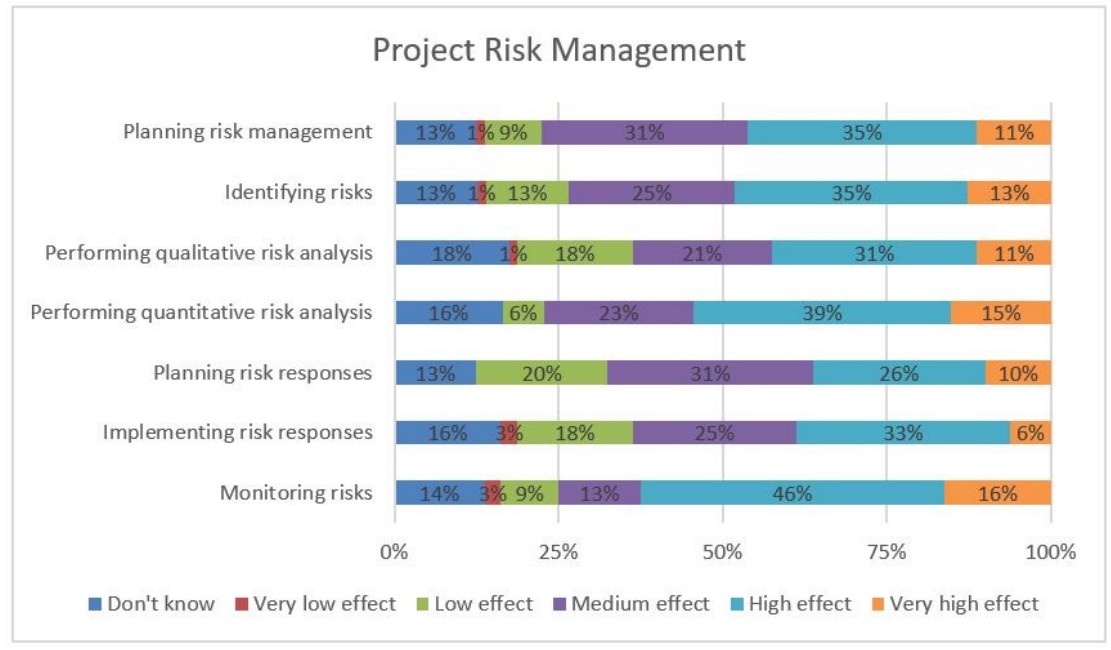

**Figure 8.** AI effect on processes of the Project Risk Management.

There was no significant correlation between how much effect respondents thought AI would have on processes of the project risk management and their age.

There was a significant correlation between how much effect respondents thought AI would have on monitoring risks and how well the respondents knew the 10 knowledge areas, rs = 0.306 and *p* = 0.031. There was a correlation between knowing the 10 knowledge areas well and believing that AI would have a high effect on monitoring risks. Out of the respondents, 68.8% who knew the 10 knowledge areas very well thought that AI would have a high effect on monitoring risks.

### 4.10. AI Effect on Project Procurement Management

Most of the participants thought that AI would have a medium to high effect on the processes of project procurement management. The result showed that 63% of participants thought that AI would have a medium to high effect on conducting procurement, 60% on controlling procurement and 58% on planning procurement management. The results can be seen in Figure 9.

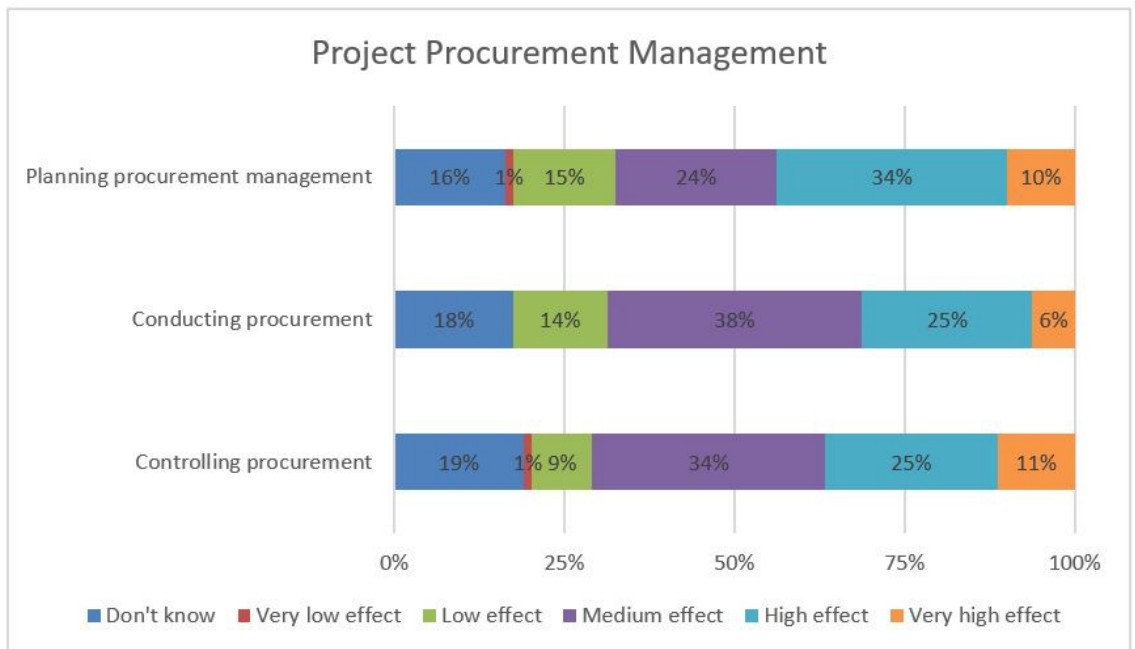

**Figure 9.** AI effect on processes of the Project Procurement Management.

There was no association between either gender or certification level and how much effect respondents believed AI would have on processes of the project procurement management. Neither was there a significant correlation between how much effect respondents thought AI would have on processes of the project procurement management and their age or their knowledge of the 10 knowledge areas.

### 4.11. AI Effect on Project Stakeholder Management

Figure 10 shows that AI will most likely have a low effect on the processes of project stakeholder management. Out of the participants, 37 to 41% estimated that AI would have a very low or low effect on identifying stakeholders, planning, and managing stakeholders' engagement. The process that AI will most likely affect the most is monitoring stakeholder engagement.

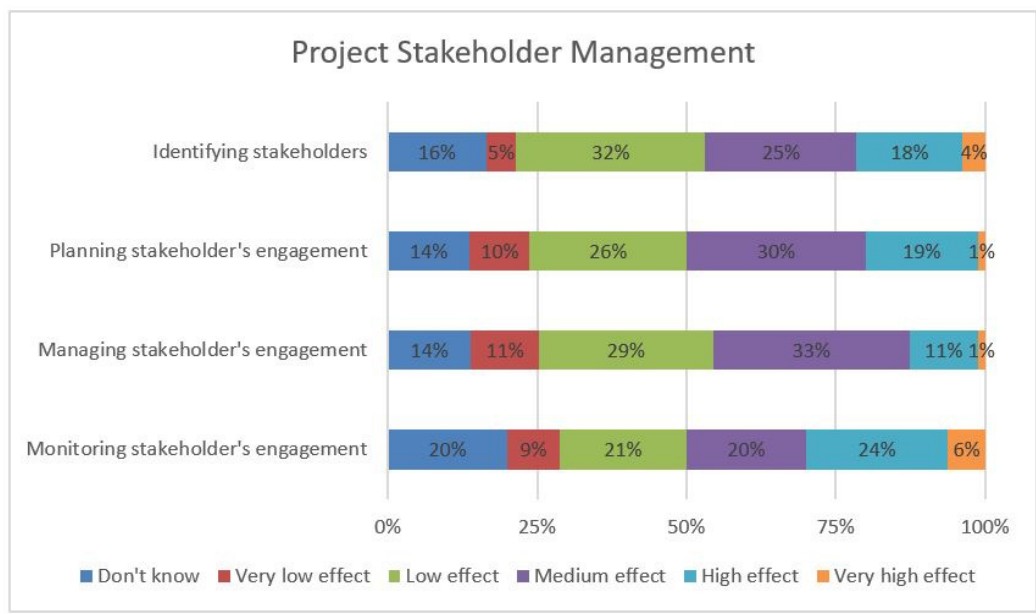

**Figure 10.** AI effect on processes of the Project Stakeholder Management.

There was no association between gender and how much effect respondents believed AI would have on processes of the project stakeholder management. There was an association between the level of certification and how much effect AI was believed to have on monitoring stakeholder engagement, *p* = 0.020. The contingency table showed that 100% of IPMA level A certified project managers thought AI would have a very low effect on monitoring stakeholder engagement. Of those who said AI would have a low effect on monitoring stakeholder engagement, 69.2% had IPMA level D certification.

There was no significant correlation between how much effect respondents thought AI would have on processes of the project stakeholder management and their age, nor their knowledge of the 10 knowledge areas.

*4.12. Background Information of Uncertain Respondents*

For each question, there was a group (12–23%) of respondents who did not know the effect that AI would have on the processes.

After the options for the questions were categorized into a binary variable, an association between the questions and background variables was analyzed. There was no association between the respondents' ages or certifications and whether they answered the main questions with "don't know."

The Pearson's chi-square test showed that there was an association between gender and answers to several of the processes of the 10 knowledge areas. The coefficient Phi ($\varphi$) that is used for $2 \times 2$ tables showed that the association was moderated for all of the associations [35]. Of those who did not know whether AI would affect developing a project charter, 75% were females, $\chi2(1, n = 80) = 4.28$, $p = 0.039$, $\varphi = 0.23$. 23% of respondents did not know how much effect AI would have on sequencing activities, and of them 72.2% were females, $\chi2 (1, n = 80) = 5.7$, $p = 0.017$, $\varphi = 0.27$. The same was for planning communication management, where 81.8% of the "don't know" answers were from females, $\chi2 (1, n = 79) = 5.82$, $p = 0.016$, $\varphi = 0.27$. There was an association between gender and all of the processes of project procurement management. Out of the respondents, 76.9% that did not know how much effect AI would have on planning procurement management were females, $\chi2 (1, n = 80) = 5.39$, $p \leq 0.02$, $\varphi = 0.26$, 78.6% for conducting procurement, $\chi2(1, n = 80) = 6.57$, $p = 0.01$, $\varphi = 0.29$, and 73.3% for controlling procurement, $\chi2 (1, n = 80) = 4.72$, $p = 0.03$, $\varphi = 0.24$. Of those who did not know whether AI would affect the process identifying stakeholder, 76.9% were females, $\chi2(1, n = 80) = 5.18$, $p = 0.023$ and $\varphi = 0.26$.

The association between how well the respondents knew the 10 project management knowledge areas and answering the main questions with "don't know" was analyzed with a contingency table and Fisher's exact test. Less knowledge of the 10 project management knowledge areas is associated with not knowing how much effect AI will have on the processes with $p < 0.05$ for all the processes except four. The processes that are not associated with a knowledge of the 10 knowledge areas were directing and managing project work, managing quality, planning communication management, and monitoring stakeholder engagement.

## 5. Discussion

The purpose of the research is to get an idea of how AI is expected to affect project management in the next 10 years. More specifically, to investigate the likely effect AI will have on the 10 PMBOK knowledge areas of project management, what processes of the 10 knowledge areas will be strongly influenced by AI, and what processes will be less affected by AI.

The research received 81 complete answers from graduated MPM students, and of the respondents, 97.6% have a project management certification, most of them from the IPMA. A larger sample group would have been preferred, which would have increased the likelihood of more complete responses. There is a strong privacy policy in Iceland that limited available resources.

The limitations of this research should be pointed out before going further. The accumulated data is in fact a set of subjective opinions, and the cognitive limitations of each and every participant in the survey may, of course, influence the outcome. Furthermore, the general limitations of on-line surveys as an instrument for data gathering should be kept in mind; there is no absolute certainty that the individuals that participated in the survey are in fact samples of the target population, or even that their answers are sincere. Based on the knowledge of this particular convenience sample, these uncertainties are deemed quite limited. Finally, it should also be pointed out that all participants in the survey are Icelandic. However, it has been shown that the importance of projects in the economy of Iceland is comparable to other Western countries [12]. Moreover, in a study by [12] it was consolidated that understanding of the principles of project management is identical in the three Western countries, Iceland, Norway, and Germany, and furthermore, that the professional education and training shared by all participants in the survey is international. As students, they have read international books on the discipline, and their knowledge has been measured against an international project management certification system [38]. As a consequence, there should be no local bias in the general outcome of this research.

The research delivered new insights and gives an idea of the potential influence of AI on the discipline of project management in the near future. The results can be useful to help the project management profession to prepare for the future, where changes in the project management work environment, skill requirements, and competencies can be expected. Knowledge and preparation for these changes can be essential to help make the most of the opportunities that AI offers. In this context it is interesting to study Table 1, which exposes the gap in the literature review concerning AI and project management.

The research indicated that of the 10 PMBOK areas of project management, project cost management, project schedule management, and project risk management are likely to benefit most from AI. This is evident when viewing the average results for the knowledge areas, also when reviewing the processes within each area. Each process of project cost management got a high effect result, where more than half of the responses indicated a very high or high effect. The process of estimating cost is the most highly affected by AI of all the 49 processes, where 21% predict a very high effect and 44% a high effect. AI will also have a very high to high effect on controlling costs, controlling schedules, and monitoring risks. This is consistent with literature dating back to 1988, where Foster talked about how AI could be used for scheduling and estimating time, cost, and risks [23] It

is also compatible with research results on AI methods that have been tested for project cost estimation, project risks estimation, and schedule success [27,30,31]. A number of AI methods have shown success in estimating project duration. This fits the result of this study, which showed AI's high effect on the processes estimating activity duration and developing a schedule, which are the processes that give start and finish dates for each activity and the project as a whole [19]. Other processes on which AI will have a high effect are creating WBS, planning procurement management, and monitoring and controlling project work.

When looking at which of the 49 processes are least affected by AI, it turns out that the analysis confirms the same as the literature—that AI is not considered to support leadership and cognitive skills [13,26]. The two processes that will be least affected by AI are developing and managing a team. In these processes, the project manager coaches and motivates his team, keeps track of their performance, gives them feedback, and mediates when problems arise [19]. The average results for the knowledge areas indicate that project stakeholder management will be least affected by AI in the next 10 years. Within that knowledge area, AI will have the lowest effect on managing stakeholder engagement, which is the process of collaborating with stakeholders to meet their requirements, expectations, and concerns [19]. These results promote the theory that there will always be a need for project managers, and human skills will possibly be more valuable in the project manager's work in the future [25,26]. AI can support the project manager's work, leading to more productivity, and giving him more time to carry out other tasks that AI cannot assist with, and more time to focus on supporting team members [19,33].

Other areas of knowledge and processes do not have a definitive high or low outcome, but are more dispersed. AI will have a medium to high effect on project quality management and project procurement management. For the other knowledge areas, the responses are distributed between low, medium, and high. A significant correlation or association is not often found between background information and the participants' responses in this research. It is more appropriate to review associations between variables when the sample size is larger. Assumptions for Pearson's chi-square test were not met because the sample size was small and frequencies in each cell were not high enough. Pearson's chi-square test gives further information than just the significance. Phi and Cramer's V can be used to present the measure of the association [35].

The data suggested that around 12 to 23% of project managers do not know if AI will have an effect on the processes of the 10 project management knowledge areas in the next 10 years. This result leads to the observation that there is possibly a lack of knowledge regarding AI in project management. Future studies should research project managers' knowledge of AI. It would have been interesting to know how well participants knew AI and how that associates with the responses. The research demonstrates an association between a lack of knowledge in the 10 knowledge areas of the PMBOK and the likelihood of not knowing how much effect AI has on the 10 knowledge areas. The results indicated that females are more likely than males to respond to certain questions with "don't know."

Taking the research further will be interesting. This will include conducting in-depth interviews with several of those who answered the research survey. A primary analysis of the survey responses can be used to design questions for the next step interviews, to get a deeper understanding of the topics, and get the respondents' perspective and thoughts on AI's future effect on project management, both positive and negative.

## 6. Conclusions

AI is becoming more and more prominent in organizations, industries and society. There is an increased discussion on its possible effects on, for instance, the future labor markets. AI is likely to increase the productivity of some employees but could replace others. Project management will not be excluded from this development and, therefore, it is interesting to speculate on what are the potential effects AI is likely to have on the profession in the future.

The aim of this research was to answer the question: How much will AI affect the 10 project management knowledge areas of PMBOK in the next 10 years? The results clearly showed that AI has a future in project management and individuals with MPM degrees in Iceland believe that AI will have a considerable effect on the project management knowledge areas in the near future. The intensity of these effects varies between knowledge areas and also between processes within each knowledge area. Project cost management, project schedule management, and project risk management are the knowledge areas that AI will have the highest effect on. AI will also have a high effect on creating WBS, planning procurement management, and monitoring and controlling project work. This shows that AI is useful for processes where historical data is used for estimation and planning, and also that AI can monitor schedules, adjust forecasts, and maintain baselines.

This research shows that knowledge areas and processes that require human leadership skills will be least affected by AI. Managing stakeholder expectations involves challenging project management tasks and requires soft skills, such as emotional intelligence. AI will have a very low effect on the processes of developing and managing teams. As with project stakeholder management, these processes require soft skills to lead people. According to the findings, AI will not perform tasks of project management that require human understanding, empathy and personal interactions. The research also indicates that some project managers find it difficult to predict the future effect of AI on project management.

There is still a lack of scientific research in the field of AI regarding project management. Scientific articles have been written about the impact of artificial intelligence on finance, industrial processes, the economy, the labor markets and society as a whole. It is interesting to examine AI usage for project management today. What kind of effect will AI have on project management in the future, and what does this mean for project managers? In addition, is it necessary to do a deeper research of project managers' understanding of AI, and how willing project management professionals are to reinvent themselves in the face of future changes?

This research will contribute to the project management profession´s interest in AI and encourage further research on the topic, so as to acquire sound knowledge on this important topic for the future. The project management profession can leverage AI by embracing the changes, working with the machines, and by nurturing the human skills and competences of the profession. The next steps are to gain further understanding on how, and to what extent, the technical and social properties of the particular knowledge areas will be affected by AI, by linking expertise in project management with expertise in artificial intelligence. Explicitly, it will be interesting to investigate in this context the three knowledge areas that this study indicates as the ones most augmented by AI. The progression of this study will reveal areas of interest on how the profession of project management will sustain itself as automation and augmentation by machine learning escalates within the discipline.

**Author Contributions:** T.V.F. and H.T.I. worked on conceptual and methodology as well as supervising the research. H.I.J. added to writing and validation. H.J. contributed to the analysis and data curation. All authors have read and agreed to the published version of the manuscript.

**Funding:** This research received no external funding.

**Institutional Review Board Statement:** Not applicable.

**Informed Consent Statement:** Not applicable.

**Data Availability Statement:** Data is contained within the article. The data was obtained as described in the article.

**Conflicts of Interest:** The authors declare no conflict of interest.

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
