# Peer review of "An Authoritative Study on the Near Future Effect of Artificial Intelligence on Project Management Knowledge Areas"

_sustainability, doi:10.3390/su13042345_

Round 1

Reviewer 1 Report

The topic of the paper deals with an important field - project management (PM) - and a very popular topic  - AI. Therefore, the paper has potential to bring some attention of scientific and professional world. But unfortunately the paper does not live to any quality standards.

This paper has some major flaws. Firstly and most important, I do not see any way that the topic of the paper fits into Sustainability journal.

The paper has a very poor structure beginning with unusual Introduction section. The literature review of PMBOK Knowledge areas and AI is very brief, which is also true for the most important part of literature review and theoretical background AI in PM. This poorly performed theoretical background does not lead to any real research questions or hypothesis. 

The aim of the paper is to get an insight into the role of AI on project management in the future. To do that, the authors have decided to randomly select master graduates in PM, expecting that they are all true experts in PM (regardless of their experiences). On top of that, there is absolutely no proof that they are experts in AI, therefore it is expected they are not in position to estimate the role of AI in PM in the future. Based on that no relevant conclusions can be made. If you want to produce relevant results, I suggest you a diiferent approach, e. g. Delph study with carefully selected experts in PM and AI. 

The results have no scientific merit. You just present the "average" opinion of random respondents on individual aspects in all 10 PMI knowledge areas. You control them with age and gender of the respondents and their subjective assessment of their expertise in PM. There is no explanation why this control variables even metter and I find no reason why you did what you did. The results do not bring to any in-depth discussion or scientific contribition.

This paper needs a lot of additional work.  

Author Response

Kindly note that the paper has been proof-read by a certified proof-reader, Mr. Simon Vaughan.

The responses are arranged in the same order as the they come in the comments and suggestions:

The topic of the paper deals with an important field - project management (PM) - and a very popular topic  - AI. Therefore, the paper has potential to bring some attention of scientific and professional world. But unfortunately the paper does not live to any quality standards.

This paper has some major flaws. Firstly and most important, I do not see any way that the topic of the paper fits into Sustainability journal.

We have extended the introduction section to deal with this comment.
"This paper focuses on project management, as a subset of general management. The importance of projects in modern societies has been defined in fiscal terms by Schoper, Wald, Ingason and Fridgeirsson (2018) who demonstrated that the share of project related work in advanced economies is one third, and rising. This fact puts the importance of project management into context and we argue that the sustainable development of any organization is very much linked to its capability to deal with change. AI is likely to shape the discipline of project managment and thus play a role in this development in the near future." We have also argued in a more comphrensive way why this strategy to use PMI standards as the platform for research is viable in this type of study.

The paper has a very poor structure beginning with unusual Introduction section. The literature review of PMBOK Knowledge areas and AI is very brief, which is also true for the most important part of literature review and theoretical background AI in PM. This poorly performed theoretical background does not lead to any real research questions or hypothesis

We agree that the structure was not clear, we have improved the structure and e.g. added numbering to all sections. We have also deleted text on project managers that was a bit out of scope of the main text.

The aim of the paper is to get an insight into the role of AI on project management in the future. To do that, the authors have decided to randomly select master graduates in PM, expecting that they are all true experts in PM (regardless of their experiences). On top of that, there is absolutely no proof that they are experts in AI, therefore it is expected they are not in position to estimate the role of AI in PM in the future. Based on that no relevant conclusions can be made. If you want to produce relevant results, I suggest you a diiferent approach, e. g. Delph study with carefully selected experts in PM and AI. 

We have defined our conveniance sample more clearly and argued why this sample can be trusted to have a certain basic understanding of the knowledge, skills and competences in project management. According to information from the MPM studies on the occupation on post graduate students 90% of them work as professional PM´s. The sample should therefore definitely contain PM experience and knowledge. However, extensive knowledge on the bolts and nuts of AI is a weakness in the sample, as is pointed out in the paper. This limitation will be addressed in a study that is planned in the spring of 2021.
A Delphy study and/or focus groups is a natural next step in this research.

The results have no scientific merit. You just present the "average" opinion of random respondents on individual aspects in all 10 PMI knowledge areas. You control them with age and gender of the respondents and their subjective assessment of their expertise in PM. There is no explanation why this control variables even metter and I find no reason why you did what you did. The results do not bring to any in-depth discussion or scientific contribition.

We accept this criticism, but we humbly object to it as it is our firm believe that the results have an errand to the community as only few publications have addressed the topic. Furthermore, this study paves the way for better understanding of the development under screening by more research.

Reviewer 2 Report

The article provides an interesting exploration into the potential effect of AI on project management in general and the project manager in general. It is competently written and well-polished. At the same time, the article also suffers from several important flaws, which are articulated below:

  1. The exact contribution of this article remains unclear. This is due to several reasons: 
    1. the introduction does not (implicitly or explicitly) specify the research gap this article aims to fill. It merely introduces the topic of AI and project management. It would help the article if the introduction already includes more insight in what we do know, or if little insight is available, how this article has dealt with the traditional problems that have caused this lack in insight;
    2. the article is ambiguous in its focus on project management, the project manager, competences and knowledge areas. What is the unit of analysis? 
    3. the review of the literature in the Sections 'PMBOK Knowledge Areas' to 'Future Project Manager' appears relatively unstructured and little systematic. As a consequence, it is difficult to understand the current state of the art in the literature on the impact of IA on the project management knowledge areas and thereby what this article contributes to what we know already. Reviewing existing literature regarding the impact of IA of the specific knowledge areas - as this is also the structure of the results - would help here.
    4. The discussion remains very close to the original results and appear under theorized. What insights are acquired and more imporant: how can these be related to existing insights?
    5. The conclusion basically ends with the same thesis as the introduction: that it is interesting and relevant to investigate IA in relation to project management. I would much more like to see a discussion on what we have gained and what further specific research questions remain.
  2. In relation to the above, it is unclear how and why the article would relate to the readership of Sustainability. Which prominent issues are discussed here, to which this article provides new or additional insights?
  3. The research aim, research question(s), questions in the study and presentation of the results appear incongruent: is it the impact or expected of IA, and is it what the impact will be, or the extent to which IA might have impact, that is investigated is discussed differently at various locations. In other words, where, what and how AI will impact seems convoluted.
  4. The article focuses only focuses on Iceland. This is not a problem in itself, but it does warrant a discussion about the extent to which Iceland can be seen as representative for the project management discipline. At minimum, a reflection about this important limitation should be included in the discussion on the limitations of the study.

Author Response

Please note that the paper has now been proof-read by a certified proofreader, Mr. Simon Vaughan.

The exact contribution of this article remains unclear. This is due to several reasons: 

  1. the introduction does not (implicitly or explicitly) specify the research gap this article aims to fill. It merely introduces the topic of AI and project management. It would help the article if the introduction already includes more insight in what we do know, or if little insight is available, how this article has dealt with the traditional problems that have caused this lack in insight;
  2. the article is ambiguous in its focus on project management, the project manager, competences and knowledge areas. What is the unit of analysis? 
  3. the review of the literature in the Sections 'PMBOK Knowledge Areas' to 'Future Project Manager' appears relatively unstructured and little systematic. As a consequence, it is difficult to understand the current state of the art in the literature on the impact of IA on the project management knowledge areas and thereby what this article contributes to what we know already. Reviewing existing literature regarding the impact of IA of the specific knowledge areas - as this is also the structure of the results - would help here.
  4. The discussion remains very close to the original results and appear under theorized. What insights are acquired and more imporant: how can these be related to existing insights?
  5. The conclusion basically ends with the same thesis as the introduction: that it is interesting and relevant to investigate IA in relation to project management. I would much more like to see a discussion on what we have gained and what further specific research questions remain.

We have extended the introduction section to deal with this comment.
"This paper focuses on project management, as a subset of general management. The importance of projects in modern societies has been defined in fiscal terms by Schoper, Wald, Ingason and Fridgeirsson (2018) who demonstrated that the share of project related work in advanced economies is one third, and rising. This fact puts the importance of project management into context and we argue that the sustainable development of any organization is very much linked to its capability to deal with change. AI is likely to shape the discipline of project managment and thus play a role in this development in the near future." We have also argued in a more comphrensive way why this strategy to use PMI standards as the platform for research is viable in this type of study.

We agree that the structure was not clear, enough we have improved the structure and e.g. added numbering to all sections. We have also deleted text on project managers that was a bit out of scope of the main text.

We have added a comment on the limited availability of published research on the potential impact of AI on the different knowledge areas of project management.

We have rephrased the research question:
"What aspects of project management, as defined by the Project Management Body of Knowledge, will be affected by artificial inteillgence in the next ten years? Detailed speculations about the essence and extent of this impact are however not within the scope of this paper."
Furthermore, we try to clarify our research focus throughout the paper to minimize this inconsistency.

In relation to the above, it is unclear how and why the article would relate to the readership of Sustainability. Which prominent issues are discussed here, to which this article provides new or additional insights? 

The article is ambiguous in its focus on project management, the project manager, competences and knowledge areas. What is the unit of analysis?

We think that this study has a value in context of sustainability as several academics and professionals are asking exactly on how the profession will evolve and sustain itself. We have tried to clarify this in the text. Regarding the methods (units of analysis) we have added a comment on this:
"The unit of analysis for this research is the discipline of project management, as defined by the ten knowledge areas of the PMBOK."

The research aim, research question(s), questions in the study and presentation of the results appear incongruent: is it the impact or expected of IA, and is it what the impact will be, or the extent to which IA might have impact, that is investigated is discussed differently at various locations. In other words, where, what and how AI will impact seems convoluted

We have added a comment on the limited availability of published research on the potential impact of AI on the different knowledge areas of project management. We have rephrased the research question:
"What aspects of project management, as defined by the Project Management Body of Knowledge, will be affected by artificial intellgence in the next ten years? Detailed speculations about the essence and extent of this impact are however not within the scope of this paper."
Furthermore, we try to clarify our research focus throughout the paper to minimize this incronsistancy.

We have added an overview of the observed links between reported impact of AI and the different PMBok knowledge areas. However, not much litterature is currently available on how AI will augment project management which, in our opinion, consolidates that this study is relavant both by it own rights and as a platform for further studies.

The article focuses only focuses on Iceland. This is not a problem in itself, but it does warrant a discussion about the extent to which Iceland can be seen as representative for the project management discipline. At minimum, a reflection about this important limitation should be included in the discussion on the limitations of the study.

We have added a comment on this in the discussion section:
"Finally, it should also be pointed out that all participants in the survey are Icelandic. However, it has been shown that the importance of projects in the economy of Iceland is comparable to other western countries (Schoper et al, 2018). Furthermore that the professional education and training shared by all participants in the survey is international, as students they have read international books on the discipline and their knowledge has been measured against an international project management certification system (Ingason and Jonasson, 2015). As a consequence, there should be no local bias in the general outcome of this research.

Reviewer 3 Report

Dear Authors,

Thank you for writing this article and it was nice to read about artificial intelligence on project management. There are some minor recommendations below.

  1. In the sentences
  • “The research was designed in an accordance the framework of the PMBOK. The sample of experts was defined based on their educational background and how likely it would be to reach out to them and commit them to take part in the survey. The sample chosen was hence individuals that had graduated with a Master in Project Management (MPM) from two Icelandic universities in the period of 2007-2019. These people were both suitable for the survey due to their knowledge and they were easy to access (Kumar, 2014)”;
  • “The purpose of this research is to get a project management specialist perspective on AI's future effect on project management, in order to get an understanding of what knowledge areas of project management will most likely in foreseeable future be affected by AI. The research was exploratory since it investigates a topic that little data is available on and where there is still a lack of scientific research and theoretical literature. The study was also prospective as it focused on the possible effect that AI could have on project management in the foreseeable future (Kumar, 2014)”

the reference to “Kumar, 2014” is unclear.

  1. ‘Quality of Research’ is based on the authors' confidence that “validity in research has to do with the appropriateness and accuracy of the research instrument used to measures a given reality. The design of the questionnaire was conducted in such a fashion that it would give sound results and accurately answer to the research question. The ten project management knowledge areas from the PMBOK were used as the research framework. The research is considered valid because each question had a logical link with an objective and the questionnaire covered all aspects needed.”

Such authors’ confidence cannot be refuted or criticized. Furthermore, an alternative opinion due to a lack of access to the questionnaire can’t be shown.

  1. The study is based on 81 complete answers of people, who had graduated with a Master in Project Management from two Icelandic universities in the period of 2007-2019.

With all due respect and trust in Icelandic universities, simply holding a degree and an international project management certification does not guarantee a high level of expertise in the assessment of the effects of artificial intelligence on project management knowledge areas in the near future. Simply holding a degree and a certificate does not even guarantee that these qualifications’ holders work in the field of project management.

The authors have most likely chosen a worthy group of experts and I have no reason to believe they did not take this into account, but this is not indicated in the work.

In any case, a set of subjective opinions does not lead to an objective assessment, if only because everybody has individual cognitive limitations. We should bear the contradictions associated with the basic principle of agnosticism in mind: it is impossible to obtain certainty based solely on subjective judgments. In addition, there is no certainty that 395 (81) emails turned out to be correct, corresponding to the individuals to whom the letters were addressed, or that, for example, their answers were sincere, that the answers were given by experts, and not by their assistants, etc.

  1. I recommend a language and stylistic editing (“…results by effectively apply project MANGEMENT methods.”, “AI technology is based on build a knowledge “This research WIL contribute to the interest…”,

At the same time, the research results are presented very well, accessible and, of course, this research will contribute to the interest of the project management profession in AI, encourage further research on the topic, as to acquired sound future knowledge on this important topic”.

Best regards,

Author Response

  1. In the sentences
  • “The research was designed in an accordance the framework of the PMBOK. The sample of experts was defined based on their educational background and how likely it would be to reach out to them and commit them to take part in the survey. The sample chosen was hence individuals that had graduated with a Master in Project Management (MPM) from two Icelandic universities in the period of 2007-2019. These people were both suitable for the survey due to their knowledge and they were easy to access (Kumar, 2014)”;
  • “The purpose of this research is to get a project management specialist perspective on AI's future effect on project management, in order to get an understanding of what knowledge areas of project management will most likely in foreseeable future be affected by AI. The research was exploratory since it investigates a topic that little data is available on and where there is still a lack of scientific research and theoretical literature. The study was also prospective as it focused on the possible effect that AI could have on project management in the foreseeable future (Kumar, 2014)”

the reference to “Kumar, 2014” is unclear.

The referrences to Kumar were removed in both cases, as they were not relevant.

‘Quality of Research’ is based on the authors' confidence that “validity in research has to do with the appropriateness and accuracy of the research instrument used to measures a given reality. The design of the questionnaire was conducted in such a fashion that it would give sound results and accurately answer to the research question. The ten project management knowledge areas from the PMBOK were used as the research framework. The research is considered valid because each question had a logical link with an objective and the questionnaire covered all aspects needed.”

Such authors’ confidence cannot be refuted or criticized. Furthermore, an alternative opinion due to a lack of access to the questionnaire can’t be shown.

We have rephrased the research question:
"What aspects of project management, as defined by the Project Management Body of Knowledge, will be affected by artificial intellgence in the next ten years? Detailed speculations about the essence and extent of this impact are however not within the scope of this paper."
Furthermore, we try to clarify our research focus throughout the paper to minimize this incronsistancy

We have defined our conveniance sample more clearly and argued why this sample can be trusted to have a certain basic understanding of the knowledge, skills and competences in project management. According to information from the MPM studies on the occupation on post graduate students 90% of them work as professional PM´s. The sample should therefore definitely contain PM experience and knowledge. However, extensive knowledge on the bolts and nuts of AI is a weakness in the sample, as is pointed out in the paper. This limitation will be addressed in a study that is planned in the spring of 2021.
A Delph study and/or focus groups is a natural next step in this research.

We have added an overview of the observed links between reported impact of AI and the different PMBok knowledge areas. However, not much litterature is currently available on how AI will augment project management which, in our opinion, consolidates that this study is relavant both by it own rights and as a platform for further studies.

The study is based on 81 complete answers of people, who had graduated with a Master in Project Management from two Icelandic universities in the period of 2007-2019.

With all due respect and trust in Icelandic universities, simply holding a degree and an international project management certification does not guarantee a high level of expertise in the assessment of the effects of artificial intelligence on project management knowledge areas in the near future. Simply holding a degree and a certificate does not even guarantee that these qualifications’ holders work in the field of project management.

We have added a comment on this in the discussion section:
"Finally, it should also be pointed out that all participants in the survey are Icelandic. However, it has been shown that the importance of projects in the economy of Iceland is comparable to other western countries (Schoper et al, 2018). Furthermore that the professional education and training shared by all participants in the survey is international, as students they have read international books on the discipline and their knowledge has been measured against an international project management certification system (Ingason and Jonasson, 2015). As a consequence, there should be no local bias in the general outcome of this research.

Round 2

Reviewer 1 Report

-

Author Response

From line 54 we have added a bit more depth to the importance of a more comprehensive understanding on the PM knowledge areas that arguably will be affected by AI.

From line 201. We have inserted a table and improved the literature search.

From line 516 we added a sentence referring to Table 1 elaborating on the importance of adding to the knowledge.

Reviewer 2 Report

The authors have made great progress in further focussing the article. The implemented changes have resulted in an article that is already much more clear on its findings and how these can be related to current insights (or the lack thereof). At the same time, I would like to encourage the authors in making taking their efforts one step further. I have three comments that, if taken into account, would benefit the article. 

  1. While the relevance of the topic has been improved, this still feels a bit ad hoc and superficial: IA will impact organizations and management, project are important for organizations (at least in fiscal terms), and therefore it is interesting to study the impact of IA on project management. I feel the argumentation line here would benefit from a bit more depth and sophistication.
  2. I really appreciate the newly added overview at the end of Section 2. I would convert this overview into a table and in this table not only include the literature, but, on the basis of the mentioned articles, also succinctly include the current state of the art of the literature in terms of content. This would allow the readers to gain easy access to the state of the art of the existing literature.
  3. In line with the above, reflect on this Table, when discussing the results and findings in Section 4-5. This help the readers to understand how this article adds to the existing literature (as discussed in Section 2).

Author Response

While the relevance of the topic has been improved, this still feels a bit ad hoc and superficial: IA will impact organizations and management, project are important for organizations (at least in fiscal terms), and therefore it is interesting to study the impact of IA on project management. I feel the argumentation line here would benefit from a bit more depth and sophistication.

From line 54 we have added a bit more depth to the importance of a more comprehensive understanding on the PM knowledge areas that arguably will be affected by AI.

I really appreciate the newly added overview at the end of Section 2. I would convert this overview into a table and in this table not only include the literature, but, on the basis of the mentioned articles, also succinctly include the current state of the art of the literature in terms of content. This would allow the readers to gain easy access to the state of the art of the existing literature.

From line 201. We have inserted a table and improved the literature search.

In line with the above, reflect on this Table, when discussing the results and findings in Section 4-5. This help the readers to understand how this article adds to the existing literature (as discussed in Section 2).

From line 516 we added a sentence referring to Table 1 elaborating on the importance of adding to the knowledge.

Round 3

Reviewer 1 Report

Thank you for considering the comments of all three reviewers, including mine. I believe that the paper has a lot of potential to be improved in the future, but as it brings some preliminary results on very important and modern topic - AI

Author Response

Thank you for your review. Yes, we are already working on further studies building on this one so it is important for us to publish is it and move on.